# Sociocultural environmental factors and childhood stunting: qualitative studies – a protocol for the Shared Values theme of the UKRI GCRF Action Against Stunting Hub

Benita Chinenye Odii,[1,2] Marie K Harder [ID] ,[1,3] Yanyan Huang,[1] Annabel Chapman,[3] Ndèye Marième Sougou,[4] Risatianti Kolopaking,[5,6] SubbaRao Gavaravarapu,[7] Amadou H Diallo,[8] Rita Anggorowati [ID] ,[5,9] Sylvia Fernandez Rao,[10] Claire Heffernan[11]

For numbered affiliations see end of article.

**Correspondence to**
Professor Marie K Harder; m.k.harder@brighton.ac.uk

## ABSTRACT

**Introduction** Stunting is a significant and growing global problem that is resisting scientific attempts to understand it in terms of direct nutrition-related determinants. In recent years, research included more complex, indirect and multifactorial determinants and expanded to include multisectoral and lifestyle-related approaches. The United Kingdom Research Initiative Global Challenges Research Fund's (UKRI GCRF) Action Against Stunting Hub starts on the premise that dominant factors of stunting may vary between contexts and life phases of the child. Thus, the construction of a typology of clustered factors will be more useful to design effective programmes to alleviate it. The Shared Values theme seeks to build a bottom-up holistic picture of interlinked cultural contextual factors that might contribute to child stunting locally, by first eliciting shared values of the groups closest to the problem and then enquiring about details of their relevant daily activities and practices, to reveal links between the two. We define shared values as what groups consider 'valuable, worthwhile and meaningful' to them.

**Methods and analysis** We will recruit 12–25 local stakeholder groups in each site (in India, Indonesia and Senegal) involved in children's food and early learning environments, such as mothers, fathers, grandmothers, teachers, market vendors and health workers. The WeValue InSitu process will be used to assist them to collectively elicit, negotiate and self-articulate their own shared values through exploration of shared tacit knowledge. Focus group discussions held immediately subsequently will ask about daily activities relevant to the children's environment. These contain many examples of cultural contextual factors potentially influencing stunting locally, and intrinsically linked to shared values articulated in the previous session.

## INTRODUCTION

Stunting was estimated to affect approximately 149 million children under the age of 5 years, mostly living in low-income and middle-income countries (LMICs),[1 2] with associated development outcomes including low economic productivity and adverse maternal reproductive issues.[1 3 4]

Research on understanding determinants has focused on direct nutrition-related determinants (such as poor maternal health and nutrition,[5] household food insecurity,[6 7] poor water, sanitation, hygiene, non-ideal breastfeeding and inappropriate complementary feeding practices[8]) strategies based on multisectoral and multifactorial approaches,[9–11] as well as exploring how different community resources, capacities and strategies could be used to mitigate child stunting. These studies build on the premise that interventions on such determinants will directly improve stunting and linear growth retardation.[11] Research has recently widened to qualitative and mixed-method studies to explore the influences on specific nutrition-related determinants of sociocultural practices and religion,[11–13] school systems, health communication campaigns, food chains.[13] However, it has been reported that even with high

### WHAT IS ALREADY KNOWN ON THIS TOPIC

⇒ Some specific direct determinants to child stunting are known, but are insufficient for a workable model: a typology incorporating localised and contextual factors is needed.

⇒ Indirect factors that affect child stunting include culturally contextual ones, but they are difficult to capture and make tangible via deductive methods.

⇒ Knowledge of *linkages* between cultural contexts and specific determinants would allow better understanding, modelling and effective interventions, but tracing them individually is resource intensive.

**WHAT THIS STUDY ADDS**

⇒ Crystallising articulated shared values of local groups—involved with children's food and learning practices—will provide broad contextual cultural understanding of those practices.

⇒ Cultural understanding around child food and learning practices may reveal grounded determinants for child stunting, with local and potentially transferable applications.

⇒ After values crystallisation, focus group discussions of lived practices around children's food and learning reveal cultural linkages with potential (including grounded) determinants of child stunting.

HOW THIS STUDY MIGHT AFFECT RESEARCH, PRACTICE OR POLICY

⇒ At a local level, this study will allow researchers to better understand local context and take it into account for socially appropriate research and intervention design.

⇒ Across the three-country studies, there will be patterns of where social context is particularly influential on stunting and the pathways for it: this will build a more accurate typology.

⇒ The novel approach for what is effectively an 'accelerated quasianthropology' will likely be well demonstrated and then ready for uptake in general development projects globally.

coverage (90%) of the 'top 10' recommended 'nutrition-specific' interventions, the stunting burden in LMICs will likely only reduce by 20%.[14 15]

We posit that instead of identifying and trying to model an increasing number of candidate factors and then linking them, a more authentic and holistic ethnographic approach could be useful in studying local shared values, which permeate local life and underpin the local cultures in which stunting is found. We define shared values as those things that groups of people consider 'valuable, worthwhile and meaningful' to them. Local life practices are built on these to some extent, including those which produce stunting. For example, in certain settings, moderate undernutrition is perceived not as a health problem, but rather a 'seasonal weight loss'.[13 15] A lack of understanding of such locally situated perceptions[16] can cause 'the creation of solutions that are neither meaningful nor beneficial to those in need', (Harder *et al*, p509)[17] whereas an understanding of underlying shared values of groups associated with children's food and education might reveal grounded information on linkages between them and direct determinants of stunting.

Therefore, the aim of the studies in the Shared Values theme of the UKRI GCRF Action Against Stunting Hub is to elicit from local populations clear indications of the in situ shared values of stakeholder groups, which are known to influence children's food and education environments, and linkages from those shared values to their relevant daily practices. To do this, we will perform qualitative research using a grounded values elicitation approach called WeValue InSitu[18 19] combined with specialised subsequent focus group discussions (FGDs), which are named Perspective EXplorations (PEX:FGD) (see figure 1). As the Action Against Stunting project is an international and interdisciplinary research team, each workstream is providing a published protocol to ensure clarity and transparency around the data produced and ensure the project is communicated to the wider research community.

WeValue InSitu was originally developed by the EU ESDinds Project[20] to make tangible the values dimension, which interlinks the social, financial and environmental pillars of sustainable development[16] and in particular to produce locally valid values-based indicators for civil society groups in Europe, South America and Mexico[17 21];

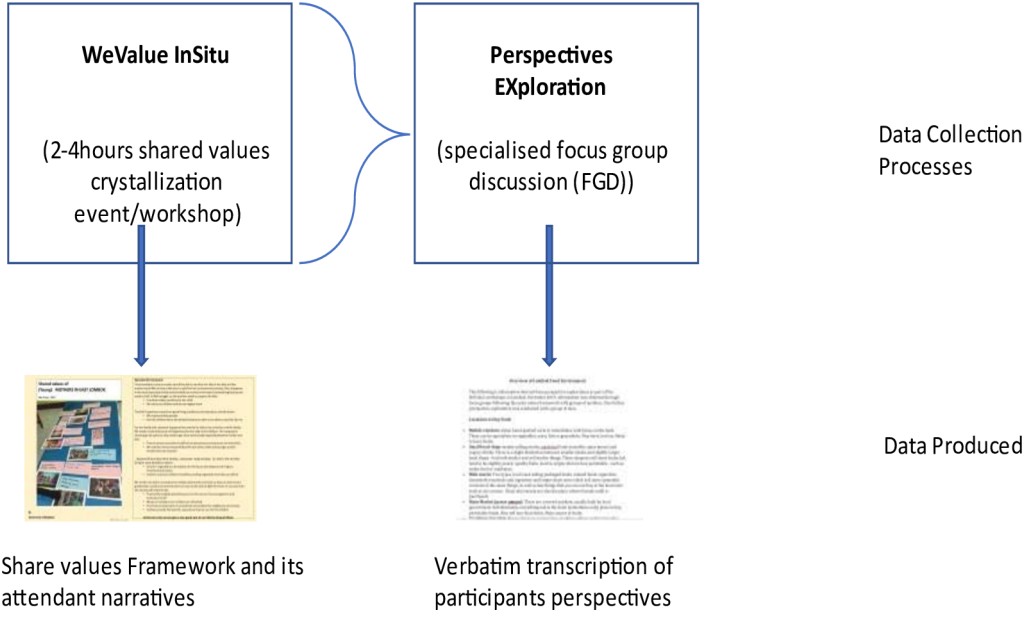

**Figure 1** Use of the WeValue InSitu approach to elicit culturally contextual factors of stunting by combining it with immediately subsequent specialised focus group discussions for Perspectives Explorations (PEX) (this should show a WeValue (WV) box and a PEX box, possibly with an arrow each to the outcomes of each: Fr+Narr and PEX Transcription. It might be nice to have a 'lens' in between the two main boxes to show the PEX is in the lens of the WV). WV, We Value.

**WeValue InSitu Process outcome: Shared values framework and narratives: "it is important to us that.."**

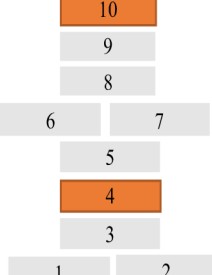

Village 7 Development Group

First of all, there should be Unity. Without unity, you can't achieve anything. That is first and foremost,

 7 …we are united to achieve our works.

Then we learn lessons from our mistakes so that we do not repeat the same mistakes,
We believe that people should contribute to the organization and not just take from it. It helps things to move well,

 8 …we learn from our mistakes

 9 …everyone contributes to the group and not just receive

Agriculture is our major means of livelihood. We should be able to market our agricultural products and get some money from that,

 10 … Agriculture is very important to us

Then we can use the money we make from selling our farm produce to develop our community,

 4 …we are able to market our farm produce.

 5 …we have development like Estates in our village

 6 …we have factories in our village. It creates employment and reduces vices

As a union, when we get any report, we must carry out proper investigation because if we don't do so, it will create disunity in the village.
And once that happens, it will cause serious damages and there will be no form of progress in the community,

 1 …We value proper investigation of information before judgment

It is important that we train our youth who will take over from us, we teach them so that when we are not there, they can take up the responsibility,

 2 …We value nurturing our youth to take over from us.

 3 …Everyone has his own building (house)

**Figure 2** An example of the output of a WeValue InSitu session: a grounded framework of statements of shared values of "what is important to us" and an accompanying narrative to introduce it.

later extended to rigorously bring local values into climate adaptation planning in Botswana[22] and land regeneration plans in Nigeria.[23]

Formal descriptions of the design-based WeValue InSitu values elicitation subprocesses are still being developed[24 25] but they involve cycles of meaning-making related to Polanyi's Personal Learning Theory, whereby the facilitator leads the group to discuss their own tacit knowledge of 'what is important to us as mothers in East Lombok' with mentions of related experiences and self-comparisons of what was most meaningful about them. In so doing, the facilitator provides a scaffolding process for the participants to learn to articulate more explicitly their tacit knowledge[25]—which is explored and negotiated as a group through critical reflections.[24 25] Each final articulated statement of 'what is important to us' is put on the table and then participants finally link them into a framework, and give a narrative description to introduce it to strangers (see figure 2). This approach always requires a specially trained facilitator, and an indigenous researcher sufficiently trained to enable delivery in the local language.

For the Stunting Hub work, WeValue InSitu (WVIS) will be used with each separate group (eg, mothers, teachers) not to produce indicators from the Frameworks, but to allow *the process* to deeply ground them in their own self-identified shared values. An immediately subsequent PEX:FGD will then suggest open topics for discussion relating to their lived roles and practices influencing children's food and learning environments, such as 'tell me how you get to market, and buy food'; 'what does the day of the children look like?'. Within these discussions many grounded, indirect contextual factors of stunting will be naturally mentioned, well-linked to each other and to underlying shared values. Analysis of the transcripts will reveal them and allow synthesis across groups to provide related cultural contextual factors for that site.

## METHODS AND ANALYSIS

The work will take place in two phases in the 5-year Action Against Stunting Hub project starting in 2019, each with 2 weeks of fieldwork visits to each site. The first phase will be at the start of the project in 2019 to provide formative research results to inform intervention designs. The second will be at the end of the project in 2023 to provide reflective feedback from the population and formative research for future work.

To build up the capacity of local facilitators, the UK expert WVIS team will first work with country lead researchers to train online about the detailed WVIS processes and to characterise the facilitators needed (practice-based facilitators with some academic background preferred). On arrival in each country, the UK team will welcome up to 20 local researchers of any seniority or experience for more detailed seminars, and

then take those who are interested through the experience of a WVIS event, followed by further seminars to understand the experience. Finally, up to six who show talent will be invited for one-to-one training during a series of WVIS events. In parallel, a simple 'certification' pathway will be set up with different levels of facilitator certification defined relative to the seminar and experiential training and reflexive exercises. A parallel pathway will be set up for WVIS analyst certification. The objective is to leave a legacy of a small independent WVIS team in each country. Concerning indigenous researchers, these should be chosen by the local research team and be capable of understanding the concept of WVIS and highly fluent in local language and English so that they can act as intermediary facilitators alongside a certified facilitator even if they themselves are not certified.

Each stakeholder group in each hub site (Kaffrine, Senegal; Hyderabad, India; East Lombok, Indonesia) is taken through the WeValue InSitu approach to articulate their envelope of shared values, (in situ and without consideration of any external topic such as stunting) and then participate in the PEX:FGD to produce descriptions of lived practices. These processes require that the group participants have lived experiences in common, and for the purpose of this stunting study those have been chosen to be stakeholder groups whose roles involve children's food or learning environments. Selection criterion thus includes both constraints, resulting in local groups of mothers, father, early year teachers, grandmothers and market vendors, with farmers and community health workers where possible. To ensure no contamination of other hub studies, these groups were chosen from adjacent residential areas of cultural similarity but not directly in the hub main cohort areas.

The WeValue InSitu Activity Stages are indicated in figure 3. In contextualisation, the group clarify their shared areas of practice and thus group context for this session. The facilitator then draws them to their tacit knowledge space by asking them to choose photos which resonate with 'what is worthwhile, meaningful and valuable' to them about their roles and presenting the ideas to the others. The trigger list contains sample statements of shared values and is used to trigger or disrupt participants to critically reflect on how their own values

might compare: this is done through individual reading and circling of resonating statements. For less-literate groups, these are read out and participants mark a Bingo card with corresponding numbers to record statements which resonate. In collective exploration, participants are asked to propose statements for discussion, and the facilitator uses techniques such as reflecting back, gentle challenging, disentangling[25] to assist the participants to compare experiences and negotiate what they collectively find 'important' about their common type of work and negotiate articulation into concise statements. When no pressing topics remain, the group negotiates the assembly of their statements into a values framework and agree a narrative for it. Workshop processes require that each participant is involved, which requires time, and produces the constraint of keeping group sizes down to 4–12 in order to keep the event within 1–4 hours. Recruitment will be by local in-country researchers, and participants offered travel reimbursements and refreshments where desired. Since this is an exploratory study aiming not at representation but for contributions to theory building of the stunting typology, only 2–3 groups of each type are initially needed for variability/saturation checks. Previous WeValue InSitu studies indicated that group sizes as small as three generally still produced sufficiently rich process and outcomes.[26]

Although we expect to be able to uncover deeply held perspectives on stunting and related topics with 12–25 workshops in each location and believe our research design to be adequate to provide new insights,[19 26] it is possible that if further detail or clarification of insights is required, then more groups would be recruited in order to reach theoretical saturation of concepts.

The PEX FGDs involved open questions about the roles of the group, which might overlap with stunting factors. For example, fathers were asked what type of learning and training they thought important for their children; grandmothers were asked about changes in food availability and different child carer types; vendors about how they sourced their produce.

The main outcomes are the PEX FGDs, to be audio-recorded and transcribed. Whether they are only transcribed in the local languages and later translated into English, or additionally translated live to produce parallel

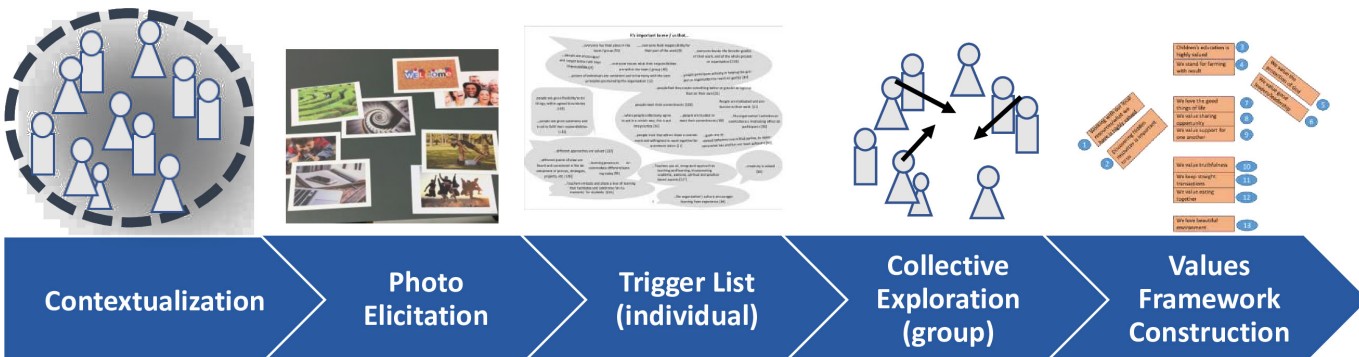

**Figure 3** The activity stages of the WeValue InSitu process.

Contextualization → Photo Elicitation → Trigger List (individual) → Collective Exploration (group) → Values Framework Construction

English audios, will depend on the final researchers involved. Meta-ethnographic translation analysis will be used,[27 28] which involves extracted clips of intended meaning, comparing these across groups, and synthesising clusters of topics (while maintaining sight of the source quotes in a grounded approach.[29] All the analysis would involve local researchers checking authenticity is not lost. The Shared Values Frameworks and narratives will be retained as descriptions of overarching cultural contexts for the FGD data.

## Patient and public involvement

Some 2–3 months before the main workshops, 6–8 local members of the public are interviewed by local researchers for material to develop the materials: the workshops' trigger statements are synthesised from those interview data. The main research itself is wholly concerned with crystallising local shared values so that they can be taken into account in the wider hub work including substudy and intervention designs. And thus informed by the priorities, experiences and preferences revealed by the WVIS approach which was, itself, codeveloped from 'action research' work alongside nine non-governmental organisations.[20] The local researchers recruit members of the public as being typical of stakeholder groups involved with children's food and learning environments. An entire later phase of 10 workshops at each site will be dedicated to presenting hub results to such stakeholders and documenting their views on its alignment with their priorities.

## DISCUSSION

This investigation is ambitious and has a risk of not producing very useful results, but at the same time has a chance of delivering a new approach for studying socioenvironmental factors of health and for designing interventions which can be more effectively implemented. It hinges on the idea that not only can shared values of groups be efficiently elicited which are grounded, holistic and in situ,[24 26] but also that they can be tentatively linked to scenarios where drivers of stunting come into play. Such candidate linkages to drivers have been successfully developed in the context of climate change perspectives[26] but it is not yet clear whether they will emerge for the more tacit practice-based scenarios where stunting might be influenced. Second, it is not clear that the researchers in the health themes of this interdisciplinary hub will be able to make use of such findings. Many disciplines are locked into specific research practices and these do not necessarily interface well with socioeconomic environmental information. Therefore, much effort will be needed both from the Shared Values team and the other teams, to investigate embedding of the information. For example, eating habits underpinned by certain shared values might provide formative research for the Food Environment team and their questionnaire development, whereas shared values showing strong community support in times of crises might be of interest to the Epigenetics team to retrospectively target stress-related indicators in parents.

The design of two separate phases of this work will be very effective in allowing examination of the explorations of the first phase to produce more mature conceptualisations of how the approach could be even more useful for informing future interventions. The second phase, at the end of the 5-year project, can thus demonstrate and possibly test new refinements of the early, more exploratory phase. It will also provide a deeply engaging cycle where feedback can be obtained from groups in the community on the processes and outcomes of the project, and thus provide data that can be presented to policy-makers, which includes representations of the public. This research space between the two spaces will allow the interdisciplinary researchers to learn more about overlaps in the diverse overlapping fields involved and consider contributions across fields. For example, what is the relevance of the applied methods of focused ethnographic studies and community-based participatory research,[30 31] or tacit–explicit knowledge transposition,[32] to the WVIS approach and do they suggest wider contributions to health research?

This study has some limitations. First, the questions that can be asked during the PEX are limited as each group will only have stamina for around an hour of PEX activity and thus if more topics are required then more groups must be recruited. Second, this approach is restricted in that it requires a trained and experienced facilitator and this requires preparation time and one-to-one in person training from the expert team. Lastly, the WVIS approach requires the facilitator to engage with local people in their own language, and this will require an indigenous researcher/translator with great skill and stamina. This researcher will also be needed to carefully go over the research conclusions reached from each FDG to ensure they are fully aligned with what was said by participants.

**Author affiliations**
¹Department of Environmental Science and Engineering, Fudan University, Shanghai, PR China
²Department of Linguistics, Igbo and Other Nigerian Languages, University of Nigeria, Nsukka, Enugu, Nigeria
³Values and Sustainability Research Group, School of Architecture, Technology and Engineering, University of Brighton, Brighton, UK
⁴Preventive Medicine and Public Health, Université Cheikh Anta Diop (UCAD), Dakar, Senegal
⁵Southeast Asian Ministers of Education Organization Regional Center for Food and Nutrition (SEAMEO RECFON)-Pusat Kajian Gizi Regional, Universitas Indonesia, Jakarta, Indonesia
⁶Faculty of Psychology, UIN Syarif Hidayatullah, Jakarta, Indonesia
⁷Nutrition Information, Communication & Health Education (NICHE) Division, ICMR-National Institute of Nutrition (NIN), Department of Health Research, Ministry of Health & Family Welfare, Govt. of India, Hyderabad, India
⁸International Research Laboratory (IRL 3189) Environnement, santé et sociétés, CNRS, UCAD, Dakar, Senegal
⁹Indonesia Creative Education Institute (ICEI), Bandung, Indonesia
¹⁰Indian Council of Medical Research, Behavioral Science Unit, Extension and Training Division, Department of Health Research, National Institute of Nutrition, Hyderabad, India

11London International Development Centre, London School of Hygiene & Tropical Medicine, London, UK

**Contributors** MKH and CH formulated the initial research question and overall study design. AC developed the specific stunting-related research question. BCO developed conceptualisation of the WeValue subprocesses. YH and AC developed appropriate data analysis techniques. RK & RA, SRG & SFR and NMS & AHD contributed to expert adaptation of the protocol for use in Indonesia, India and Senegal, respectively. BCO and MKH wrote the paper.

**Funding** This work was supported by the UKRI GCRF grant number MR/S01313X/1.

**Competing interests** None declared.

**Patient and public involvement** Patients and/or the public were involved in the design, or conduct, or reporting, or dissemination plans of this research. Refer to the Methods section for further details.

**Patient consent for publication** Not applicable.

**Ethics approval** This study involves human participants. Ethical approval for the hub was granted by the ethics committee of the London School of Hygiene and Tropical Medicine (17915/RR/17513), by the social science research ethical review board at the Royal Veterinary College (URN SR2020-0197) and by the International Livestock Research Institute Institutional Research Ethics Committee (ILRI-IREC2020-33). In-country approvals were granted by: National Institute of Nutrition (ICMR), Ministry of Health and Family Welfare, Government of India (CR/04/I/2021); Health Research Ethics Committee, University of Indonesia and Cipto Mangunkusumo Hospital (KET-887/UN2.F1/ETIK/PPM.00.02/2019); Comité National d'Ethique pour la Recherche en Santé, Senegal (Protocole SEN19/78). Ethical approval for the Shared Values theme was granted by the CEM Research Ethics Panel of the University of Brighton (2019-2177). Participants gave informed consent to participate in the study before taking part.

**Provenance and peer review** Not commissioned; externally peer reviewed.

**Data availability statement** Data are available upon reasonable request.

**ORCID iDs**
Marie K Harder http://orcid.org/0000-0002-1811-4597
Rita Anggorowati http://orcid.org/0000-0002-8279-8508

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
