## [Reviewer comments · BMJ Paediatrics Open]

ARTICLE DETAILS

TITLE (PROVISIONAL)	Sociocultural environmental factors and childhood stunting: qualitative studies- a protocol for the Shared Values theme of the UKRI-GCRF Action Against Stunting Hub
AUTHORS	Odi, Benita Harder, Marie Huang, Yanyan Chapman, Annabel Sougou, Ndèye Marième Kolopaking, Risatianti Gavaravarapu, SubbaRao Diallo, Amadou Anggorowati, Rita Fernandez Rao, Sylvia Heffernan, Claire

VERSION 1 - REVIEW

REVIEWER	Reviewer Name: Dr. Imteyaz A. Khan Institution and Country: Saint Peter's University Hospital, Pediatrics-NICU, USA
REVIEW RETURNED	16-Mar-2023

GENERAL COMMENTS	A well-written manuscript describes the proposed methodological plan and strategy for this study. Abstracts provide a brief overview of the problem and the proposed methodology. Materials and methods are explained clearly. The methodology is appropriate for the study goals. I like the concept of a holistic preventive approach to malnutrition leading to stunting. I noticed a minor typing mistake. Line 20 - Typing error "enquiriing." Authors should explain why they want to publish this manuscript before the completion of the study.
---

REVIEWER	Reviewer Name: Dr. Sunil Karande Institution and Country: Seth Gordhandas Sunderdas Medical College, India
REVIEW RETURNED	26-Mar-2023

GENERAL COMMENTS	The Authors have devised a situated values-based approach to better identify and understand sociocultural environmental factors affecting childhood stunting. The entire manuscript is essentially to describe the "protocol" of the proposed research study [the Shared Values theme of the UKRI-GCRF Action Against Stunting Hub]. My suggestions to the Authors are: (i) Please simplify the English grammar and words in the manuscript.
---

	(ii) State a cross reference for the sentence: "Previous WeValue InSitu studies indicated that group sizes as small as 3 generally still produced sufficiently rich process and outcomes". (iii) If possible - state the proposed time frame of the research study either in the text or as a figure. (iv) Write a few details of how specially trained facilitators, and indigenous researchers would be identified and trained - and how their competence would be judged across the different study sites. This would probably help in judging the scientific yield of the research study across study sites. (iv) In a brief paragraph - state clearly if the authors would alter their methodology if the yield of the proposed research study (new information / newer insights) is inadequate. (v) The proposed study would have its limitations. The authors should write a brief paragraph on this -either from their own previous experience or from known published literature.
--	---

VERSION 1 – AUTHOR RESPONSE

Editor in Chief Comments to Author:

1.1 Title shorten to "Sociocultural environmental factors and childhood stunting: qualitative studies- a protocol for the Shared Values theme of the UKRI-GCRF Action Against Stunting Hub"

RESPONSE: We have accepted the suggested shortened title

1.2 The paper needs a discussion section adding

RESPONSE: We have now included a Discussion section, including Limitations:

DISCUSSION

This investigation is ambitious and has a risk of not producing very useful results, but at the same time has a chance of delivering a new approach for studying socioenvironmental factors of health, and for designing interventions which can be more effectively implemented. It hinges on the idea that not only can shared values of groups be efficiently elicited which are grounded, holistic, and in-situ(25,27), but also that they can be tentatively linked to scenarios where drivers of stunting come into play. Such candidate linkages to drivers have been successfully developed in the context of climate change perspectives(27) but it is not yet clear whether they will emerge for the more-tacit practice-based scenarios where stunting might be influenced. Secondly, it is not clear that the researchers in the health themes of this interdisciplinary Hub will be able to make use of such findings. Many disciplines are locked into specific research practices and these do not necessarily interface well with socioeconomic environmental information. Therefore, much effort will be needed both from the Shared Values team and the other teams, to investigate embedding of the information. For example, eating habits underpinned by certain shared values might provide formative research for the Food Environment team and their questionnaire development, whereas shared values showing strong community support in times of crises might be of interest to the Epigenetics team to retrospectively target stress-related indicators in parents.

The design of two separate phases of this work will be very effective in allowing examination of the explorations of the first phase to produce more mature conceptualizations of how the approach could be even more useful for informing future interventions. The second phase, at the end of the 5-year project, can thus demonstrate and possibly test new refinements of the early, more exploratory phase. It will also provide a deeply engaging cycle where feedback can be obtained from groups in the community on the processes and outcomes of the project, and thus provide data that can be presented to policy-makers which includes representations of the public. This research space

between the two spaces will allow the interdisciplinary researchers to learn more about overlaps in the diverse overlapping fields involved, and consider contributions across fields. For example, what is the relevance of the applied methods of Focused Ethnographic Studies (FES), and Community Based Participatory Research (CBPR)(31,32), or tacit-explicit knowledge transposition(33), to the WVIS approach, and do they suggest wider contributions to health research?

2. Reviewer: 1

Dr. Imteyaz Khan, Saint Peter's University Hospital, Rutgers Robert Wood Johnson Medical School

Comments to the Author

A well-written manuscript describes the proposed methodological plan and strategy for this study.

Abstracts provide a brief overview of the problem and the proposed methodology. Materials and methods are explained clearly.

The methodology is appropriate for the study goals. I like the concept of a holistic preventive approach to malnutrition leading to stunting.

2.1 I noticed a minor typing mistake. Line 20 - Typing error "enquiriing."

RESPONSE: this is now corrected.

2.2 Authors should explain why they want to publish this manuscript before the completion of the study.

RESPONSE:

We believe that it is important to publish the protocol paper ahead of the completion of the study to ensure that the research community is aware of the work of the Action Against Stunting Hub as a whole and to provide clarity and transparency in our future publications of results. In light of your comment, we have added the following statement to the introduction of the paper:

"As the Action Against Stunting project is an international and interdisciplinary research team, each workstream is providing a published protocol to ensure clarity and transparency around the data produced and ensure the project is communicated to the wider research community"

Reviewer: 2

Dr. Sunil Karande, Seth Gordhandas Sunderdas Medical College

Comments to the Author

The Authors have devised a situated values-based approach to better identify and

understand sociocultural environmental factors affecting childhood stunting. The entire manuscript is essentially to describe the "protocol" of the proposed research study [the Shared Values theme of the UKRI-GCRF Action Against Stunting Hub].

My suggestions to the Authors are:

(i) Please simplify the English grammar and words in the manuscript.

RESPONSE:

The manuscript has been carefully proofread and the grammar simplified where possible.

(ii) State a cross reference for the sentence: "Previous WeValue InSitu studies indicated that group sizes as small as 3 generally still produced sufficiently rich process and outcomes".

RESPONSE: A reference has been inserted: Yanyan Huang, Wenhao Wu, Yunshu Xue, Marie K. Harder (2022). Perceptions of climate change impacts on city life in Shanghai: Through the lens of shared values, *Cleaner Production Letters*, 3,100018,

<https://doi.org/10.1016/j.cpl.2022.100018>.

(iii) If possible - state the proposed time frame of the research study either in the text or as a figure.

RESPONSE: We have added the following statement to the paper:

The work will take place in two phases in the 5-year Action Against Stunting Hub project starting in 2019, each with two weeks of fieldwork visits to each site. The first phase will be at start of project in 2019 to provide formative research results to inform intervention designs. The second will be at end of project in 2023 to provide reflective feedback from the population and formative research for future work.

(iv) Write a few details of how specially trained facilitators, and indigenous researchers would be identified and trained - and how their competence would be judged across the different study sites. This would probably help in judging the scientific yield of the research study across study sites.

RESPONSE: We have added the following statement to the paper:

"To build up capacity of local facilitators, the UK expert WVIS team will first work with country lead researchers to train online about the detailed WVIS processes and to characterise the facilitators needed (practice-based facilitators with some academic background preferred). On arrival in each country the UK team will welcome up to 20 local researchers of any seniority or experience for more detailed seminars, and then take those who are interested through the experience of a WVIS event, followed by further seminars to understand the experience. Finally, up to 6 who show talent will be invited for 1-2-1 training during a series of WVIS events. In parallel a simple 'Certification' pathway will be set up with different levels of Facilitator Certification defined relative to the seminar and experiential training and reflexive exercises. A parallel pathway will be set up for WVIS Analyst Certification. The objective is to leave a legacy of a small independent WVIS team in each country. Concerning indigenous researchers, these should be chosen by the local research team and be capable of understanding the concept of WVIS and highly fluent in local language and English so that they can act as intermediary facilitators alongside a Certified Facilitator even if they themselves are not Certified."

(iv) In a brief paragraph - state clearly if the authors would alter their methodology if the yield of the proposed research study (new information / newer insights) is inadequate.

RESPONSE: We have added the following statement to our methods and analysis section:

"Although we expect to be able to uncover deeply held perspectives on stunting and related topics with 12-25 workshops in each location and believe our research design to be adequate to provide new insights (19,27) it is possible that if further detail or clarification of insights is required, then more groups would be recruited in order to reach theoretical saturation of concepts."

(v) The proposed study would have its limitations. The authors should write a brief paragraph on this - either from their own previous experience or from known published literature.

RESPONSE: We have added the following statement to the paper:

“This study has some limitations. Firstly, the questions that can be asked during the PEX are limited as each group will only have stamina for around an hour of PEX activity and thus if more topics are required then more groups must be recruited. Secondly, this approach is restricted in that it requires a trained and experienced facilitator and this requires preparation time and 1-2-1 in person training from the expert team. Lastly, the WVIS approach requires the Facilitator to engage with local people in their own language, and this will require an indigenous researcher/ translator with great skill and stamina. This researcher will also be needed to carefully go over the research conclusions reached from each FDG, to ensure they are fully aligned with what was said by participants.”